# Odontogenic Effect of Icariin on the Human Dental Pulp Cells

**DOI:** 10.3390/medicina58030434

**Published:** 2022-03-16

**Authors:** Guo Liu, Ying Yang, Kyung-San Min, Bin-Na Lee, Yun-Chan Hwang

**Affiliations:** 1Department of Conservative Dentistry, School of Dentistry, Dental Science Research Institute, Chonnam National University, Gwangju 61186, Korea; dentistliu@wmu.edu.cn (G.L.); bnlee13@jnu.ac.kr (B.-N.L.); 2Department of Endodontics, School and Hospital of Stomatology, Wenzhou Medical University, Wenzhou 325000, China; 3Dental Implant Center, School and Hospital of Stomatology, Wenzhou Medical University, Wenzhou 325000, China; dentistyang@wmu.edu.cn; 4Department of Conservative Dentistry, School of Dentistry, Jeonbuk National University, Jeonju 54896, Korea; endomin@gmail.com

**Keywords:** icariin, odontogenesis, mineralization, human dental pulp cell

## Abstract

*Background and Objectives*: Human dental pulp cells (HDPCs) can be used for dentin regeneration due to its odontogenic differentiation property. Icariin can induce osteogenic differentiation of stem cells. However, its potential to induce odontogenic differentiation of HDPCs remains unclear. Thus, the aim of this study was to evaluate the capacity of icariin to induce odontogenic differentiation of HDPCs and investigate the underlying molecular mechanism. *Materials and Methods*: Cell viability assay was used to detect the cytotoxicity of icariin to HDPCs. Effect of icariin on HDPCs chemotaxis was measured by scratch migration assay. The mineralized and odontogenic differentiation of HDPCs was assessed by alkaline phosphatase (ALP) staining, alizarin red S (ARS) staining, real-time PCR, and Western blot of dentin matrix protein 1 (DMP 1) and dentin sialophosphoprotein (DSPP). In addition, Mitogen-activated protein kinase (MAPK) signaling pathway of icariin-induced biomineralization was investigated by Western blot. *Results*: Cells treated with icariin at all concentrations tested maintained viability, indicating that icariin was biocompatible. Icariin accelerated HDPCs chemotaxis (*p* < 0.05). Expression levels of related odontogenic markers were increased in the presence of icariin (*p* < 0.05). Icariin-induced odontogenic differentiation occurred via activation of the MAPK signaling pathway. Furthermore, MAPK inhibitors suppressed expression levels of DSPP and DMP 1 protein, ALP activity, and mineralization of HDPCs. *Conclusions:* Icariin can upregulate odontogenic differentiation of HDPCs by triggering the MAPK signaling pathway.

## 1. Introduction

Dental pulp is a kind of vascularized and innervated tissue that could maintain homeostasis and physiological vitality of teeth by transporting oxygen and nutrients [1]. A series of serious consequences such as pulpitis, apical periodontitis, and tooth extraction could be caused by losing pulp viability after pulp exposure [2,3]. Therefore, it is necessary to keep pulp viability for the longevity of teeth.

Nowadays, direct pulp capping and pulpotomy are main therapeutic ways to preserve pulp vitality [4]. Compared with pulpotomy, direct pulp capping has attracted more attention due to its advantages of being more conservative and easier operation [5]. The prognosis of direct pulp capping depends on the biocompatibility of the material which could directly cover exposure pulp tissue to induce odontogenic differentiation of human dental pulp cells (HDPCs) and subsequently form reparative dentine [6,7]. Therefore, a pulp capping material with weak cytotoxicity and good odontogenesis ability is needed based on the understanding of repair processes of pulp tissue [8,9].

Icariin (4′-O-methyl-8-γ, γ-dimethylallylkaempferol-3-rhamnoside-7 glucoside) is flavonoid glucoside derived from the plant genus Epimedium [10]. Accumulating evidence has indicated that icariin exhibits multi functions, including promoting cell migration and proliferation [11], stimulating angiogenesis, repressing bone resorption, and promoting bone formation [10,12,13,14,15,16]. So far, it has been proposed that odontogenesis and osteogenesis share a similar dynamic procedure and mechanism in the process of the mineral phase, including cell migration, extracellular matrix synthesis, and calcium ion deposition [7,8,17,18,19,20]. Here, we evaluated the biocompatibility and effect of icariin on odontogenic capacity human dental pulp cells. We further explored its underlying molecular mechanism.

## 2. Materials and Methods

### 2.1. Sample Collection, Cell Isolation, and Cell Culture

The Institutional Review Board of Chonnam National University Dental Hospital approved this study (IRB No: CNUDH 2016-009). Written informed consents were obtained from all participants. Pulp cells of human third molars were isolated from healthy participants. Immediately after tooth extraction, the pulp tissue was separated aseptically and minced (1 mm^3^) using a surgical scissor. Pulp tissues were placed in a 60 mm dish containing α-minimum essential medium (α-MEM, Gibco Invitrogen, Grand Island, NY, USA) supplemented with 10% fetal bovine serum (FBS, Gibco Invitrogen, Grand Island, NY, USA), 100 U/mL penicillin, and 100 mg/mL streptomycin (Gibco Invitrogen, Grand Island, NY, USA) and incubated at 37 °C with 5% CO^2^. Every 3 days, the culture medium was changed. When cell confluence reached 75–80%, subculture was performed for expansion. They were passaged at a 1:3 ratio. Passages 3–5 were used in subsequent experiments.

### 2.2. Icariin Treatment

Icariin (10 mg) was purchased from Sigma (Proproof, Dutendorfer, Germany). Icariin was diluted with dimethyl sulphoxide (DMSO; Sigma-Aldrich, St. Louis, MO, USA) according to the manufacturer’s instruction to obtain concentrations of 0.1 μM, 1 μM, 10 μM, and 100 μM. Different concentrations were added to growth medium (GM) to evaluate its effect on HDPCs.

### 2.3. Cytotoxicity Assay

Cell viability assays were carried out using an Ez-Cytox Enhance cell viability assay kit (Dogen, Seoul, Korea). Briefly, 2 × 10^3^ cells/well of HDPCs were seeded in 96-well plates and incubated at 37 °C under 5% CO_2_ for 24, 48, 72 h. To avoid evaporation of the medium at the edge of the plate, Dulbecco’s buffered phosphate saline solution (DPBS; Welgene, Daegu, Korea) was placed into wells of each edge. Subsequently, icariin at different concentrations (0.1 μM, 1 μM, 10 μM, 100 μM) was added to different groups. After a period of incubation, 10 μL Ez-Cytox reagent was added to each well. After incubation for 2 h, optical densities were measured at 450 nm using a microplate spectrophotometer reader (Thermo Scientific, Waltham, MA, USA).

### 2.4. Odontogenic Differentiation

For odontogenic potential experiments, confluent cells were induced in odontogenic differentiation medium (OM) by supplementing 50 μg/mL of ascorbic acid (Sigma-Aldrich, St. Louis, MO, USA) and 10 mmol/L β-glycerophosphate (Stata Cruz Biotechnology Inc., Dallas, TX, USA) in the growth medium mentioned above. Icariin was added at 0.1 μM, 1 μM, or 10 μM to the OM. The medium was refreshed every two days.

### 2.5. Scratch Migration Assay

To detect the capability of HDPCs treated with icariin for chemotaxis, a total of 2 mL (3 × 10^5^ cells) of cells was seeded into 6-well plates. After overnight incubation, scratches were created using 200 μL pipette tips and washed twice with DPBS. To quantify wound migration area over time, images was captured at 0, 6, 12, and 24 h with a microscope (Olympus, Tokyo, Japan). Relative areas of cell migration were quantified by densitometry using ImageJ (National Institutes of Health, Bethesda, MD, USA).

### 2.6. Alizarin Red S (ARS) Staining Assay and Alkaline Phosphatase (ALP) Staining Assay

To evaluate calcium deposits, HDPCs were seed at an initial density of 2 × 10^4^ cells per well in a 48-well plate. Cells were incubated with GM and OM in the presence or absence of icariin at 0.1 μM, 1 μM, or 10 μM for 14 days. After that, cells were gently rinsed with DPBS and fixed with absolute ethanol (Emsure, Darmstadt, Germany) for 30 min at room temperature, followed by removal of absolute ethanol. Wells were allowed to dry completely. Fixed cells were dyed with 2% Alizarin red S (Lifeline Cell Technology, Fredeerick, MD, USA) for 20 min for color development. Excess stain was removed by repetitive washing with distilled water. Stained samples were scanned using an EPSON Perfection V550 Photo (EPSON, Los Alamitos, CA, USA). For quantitative analysis, calcium deposits were dissolved with 10% cetylpyridinium chloride (CPC, PH = 7.0). The absorbance of the supernatant was measured at 540 nm using a microplate spectrophotometer reader (Thermo Scientific, Waltham, MA, USA).

Alkaline phosphatase (ALP) activity was assayed after treatment with different concentrations of icariin. At 10 days after treatment, cells were fixed with 70% ice-cold ethanol and then stained with 250 μL of 1-step NBT/BCIP reagent (Thermo Fisher Scientific Inc., Rockford, IL, USA) for 30 min. Stained samples were scanned as described earlier. Stained mineralization nodules were immersed in 300 μL of 10% CPC for 30 min and the absorbance was measured at 562 nm.

### 2.7. RNA Isolation and Gene Expression Analysis by Quantitative Real-Time Polymerase Chain Reaction

The following experiments were performed to quantify mRNA expression levels of specific odonto/osteogenic differentiation genes, including dentin matrix protein 1 (*DMP 1*), dentin sialophosphoprotein (*DSPP*), runt-related transcription factor 2 (*RUNX2*), alkaline phosphatase (*ALP*), osteocalcin (*OCN*), and bone sialoprotein (*BSP*). After 24 h of incubation, cells were incubated in a 6-well plate at an initial density of 2 × 10^5^. Cells were then incubated with GM and OM in the presence or absence of 10 μM icariin for 3 and 5 days. Total RNAs were isolated from cells using Trizol (Invitrogen, Carlsbad, CA, USA). RNA concentration and purity were evaluated with a NanoDropTM 2000 spectrophotometer (Thermo Fisher Scientific, Rockford, IL, USA), followed by cDNA synthesis from 2 μg total RNA using an AccessQuickTM RT-PCR system (Promega, Madison, WI, USA). Amplification was conducted using a SYBR Green PCR Kit (Qiagen, Valencia, CA, USA) on a Rotor-Gene Q real-time PCR Cycler (Corbett Research, Sydney, Australia). Target gene expression levels were normalized with the expression of Glyceraldehyde-phosphate dehydrogenase (*GAPDH*). Sequences of primer were synthesized by Bioneer (Daejeon, Korea). They are listed in Table 1. Fold change was calculated with the 2^−ΔΔCT^ method [21].

### 2.8. Western Blot Analysis

To determine if icariin could improve the expression of odontogenic differentiation protein and activate the MAPK signaling pathway, HDPCs were seeded into 60 mm dishes with a density of 3 × 10^5^ cells/well and incubated with OM with or without icariin at 10 μM for 3 and 5 days. After stimulation, cells were lysed in protein lysis buffer (Cell Signaling Technology, Beverly, MA, USA) containing 1 mM protease inhibitor phenylmethanesulfonyl (PMSF). Protein concentration was quantified using a lower protein assay reagent (Bio-Rad Laboratories, Hercules, CA, USA). Equivalent protein samples were subjected to 10% sodium dodecyl sulphate-polyacrylamide gel electrophoresis (SDS-PAGE) and transferred to polyvinylidene difluoride membranes. Membranes were blocked with 5% skim milk dissolved in PBS containing 0.1% Tween 20 (PBST) (Biosesang, Sungnam, Korea) for 1 h. By probing the membrane with a primary antibody, the specific protein of interest was detected. Primary antibodies included anti-DSPP (Thermo Fisher Scientific, Rockford, IL, USA), anti-DMP-1(Abcam, Cambridge, UK), anti-ERK, anti-phospho-ERK, anti-JNK, anti-phosphor-JNK, anti-p38, and anti-phosphos-p38 (Cell Signaling Technology, Beverly, MA, USA). After incubating with primary antibodies at 4 °C overnight, membranes were washed with PBST and incubated with horseradish peroxidase-conjugated secondary antibodies (Sigma-Aldrich, St. Louis, MO, USA). Finally, immunoreactive bands were visualized with a Western blot analysis imaging system (Ez-capture; Atto, Tokyo, Japan).

### 2.9. Statistical Analysis

Three independent experiments were performed. All experiment data were normalized against the control and expressed by mean ± standard deviation (SD). All statistical analyses of data were performed using GraphPad Prism 8 (GraphPad software Inc., San Diego, CA, USA). Statistical differences were determined using one-way analysis of variance (AVOVA) followed by Turkey’s post. Statistical significance was accepted at *p* < 0.05.

## 3. Results

### 3.1. Effects of Icariin on Cell Viability and Migration of HDPCs

To assess cytotoxicity of icariin at different concentrations (0, 0.1, 1, 10, and 100 μM), the WST-1 assay was performed. Icariin at concentrations ranging from 0.1 to 100 μM did not affect cell viability up to 72 h (Figure 1A). Therefore, icariin at all concentrations tested was biocompatible. Its effect on HDPCs migration and odontogenesis were subsequently investigated.

To examine the migration capability of HDPCs after icariin treatment, scratch migration assay was performed. As shown in Figure 1B,C, icariin enhanced the migration of HDPCs at 6 h when compared with the GM group. After 12 h, the migration of HDPCs in the icariin treated group was significant increased over that in the OM group (* *p* < 0.05, ** *p* < 0.01, *** *p* < 0.001).

### 3.2. Icariin Accelerated ALP Activity and Mineralization Ability of HDPCs

To detect the effect of icariin on mineralization of HDPCs, we measured ALP activity and calcium nodule deposition. After 10 days of induction, icariin at all concentrations significantly improved ALP activity when compared with the GM group (Figure 2A,B) (*** *p* < 0.001). Although cells were stimulated with 0.1 or 1 μM icariin showed comparable results with the OM group, a remarkable increase of ALP staining was observed in the 10 μM icariin group (## *p* < 0.01). Based on ARS staining, mineralization was dose-dependently enhanced by icariin after 14 days of treatment when compared with the GM group (*** *p* < 0.001). Treatment with 1 μM or 10 μM icariin significantly increased mineralization compared to the OM group (Figure 2C,D) (## *p* < 0.01, and ### *p* < 0.001). Based on our results, the concentration of 10 μM icariin was selected for subsequent experiments.

### 3.3. Effect of Icariin on Gene Expression and Protein Expression of Odontogenic Markers in HDPCs

To study the impact of icariin on odontogenic differentiation of HDPCs, we examined expression levels of odontogenic marker genes such as *DMP 1*, *DSPP*, *RUNX2*, *ALP*, *OCN*, and *BSP* on day 3 and day 5. We found that icariin increased the expression levels of *DMP 1*, *DSPP*, *RUNX2*, *ALP*, *OCN*, and *BSP* mRNAs on day 3 (Figure 3A) (* *p* < 0.05, ** *p* < 0.01). Such up-regulation expression was more significant on day 5 (Figure 3B) (* *p* < 0.05, ** *p* < 0.01, *** *p* < 0.001). Western blot analysis results showed that icariin increased DSPP and DMP 1 protein levels, consistent with the above up-regulation results of genes (Figure 3C). Collectively, these findings suggest that 10 μM icariin could increase the odontogenic differentiation of HDPCs.

### 3.4. Icariin-Mediated Activation of Mitogen-Activated Protein Kinase Signaling during Odontogenic Differentiation of HDPCs

To elucidate the molecular mechanism involved in the stimulation of HDPCs differentiation by icariin, we investigated the role of mitogen-activated protein kinase (MAPK) pathway during icariin-promoted odontogenic differentiation. We measured total protein and phosphorylated protein levels of JNK, p38, and ERK using Western blot. Results showed that icariin increased phosphorylation levels of JNK, p38, and ERK within 10 min after induction (Figure 4A).

To further explore the effect of icariin on JNK, p38, and ERK signaling involved in icariin-induced enhancement of the differentiation phenotype, cells were pretreated with or without inhibitors SP600125 (JNK inhibitor), SB202190 (p38 inhibitor), and U0126 (ERK inhibitor) (Cell Signaling Technology, Beverly, MA, USA) and then treated with or without icariin. Expression levels of DSPP and DMP 1 protein (Figure 4B) were significantly suppressed by the presence of MAPK inhibitors. After 10 or 14 days, the ALP level (Figure 4C,D) and calcium deposition (Figure 4E,F) in inhibitor (SP600125, SB202190 and U0126) treated groups were significantly downregulated when compared to those in the OM and OM + icariin groups.

## 4. Discussion

Currently, icariin has gained attention due to its bone remodeling properties [10,14], anti-inflammatory effect [22], and angiogenesis-stimulating ability [23] in different fields, revealing a promising therapeutic advantage in vital pulp therapy. Despite that, to date, the role of icariin in odontogenic differentiation in HDPCs has not been reported yet. Therefore, this is the first study to provide evidence that icariin can affect the odontogenic differentiation of HDPCs.

The biocompatibility of icariin must be evaluated prior to clinical use because it is directly related to the health of dental pulp tissues. In the present study, icariin treatments at all concentrations tested were biocompatible as cell viability was maintained, although 100 μM icariin showed a tendency to decrease cell viability (not significant). This finding was different from the report by Fan [24]. Such difference might be related to different cell types and culture criteria tested.

Tertiary dentin generation is a key factor in vital pulp therapy, which depends on injured areas cell migration, early odontoblastic differentiation and late biomineralization capacity of HDPCs [9,25]. In the current study, a series of experiment results demonstrated the odontogenic effect of icariin. First, icariin significantly upregulated the chemotaxis of HDPCs based on scratch migration assay. Migration is an initial tissue healing process and repair response. When carious pulp is exposed, undifferentiated progenitor cells residing around the exposure tissue will migrate into injured sites [20]. Under the influence of medication, pulp will heal and promote reactionary dentin formation. Therefore, the ability of cells to migrate is beneficial to tissue reparation [26]. Second, DSPP and DMP-1 were two classic odontogenic differentiation markers as members of the small integrin-binding ligand N-linked glycoproteins family (SIBLINGs) [9,27,28,29]. Our results showed that 10 μM icariin enhanced mRNA and protein expression levels of *DSPP* and *DMP-1* known to play a vital role in the progression and formation of pulp-dentin complex [30,31]. Third, osteoblasts and odontoblasts share some similar gene expression, including *RUNX2*, *ALP*, *OCN*, and *BSP*. *RUNX2* and *ALP* are related to the expression of early-stage markers in bone and dentine. *RUNX2* is also a key transcription factor associated with odontogenesis [32,33]. Expression levels of *BSP* and *OCN* are essential in the late stage of mineralization and tooth development [32,34]. In this study, *RUNX2*, *ALP*, *OCN*, and *BSP* mRNA levels were obviously upregulated in HDPCs when stimulated by icariin, which further validated that icariin could elevate biomineralization and odontoblastic differentiation. Lastly, this study displayed that icariin recognizably increased odontogenesis verified by ALP activity and calcified nodule formation. These results corroborated that icariin could be considered as a novel agent for odontogenic differentiation of HDPCs.

Cytodifferentiation is an intricate dynamic process that involves the coordination of multiple signaling pathways. Cells perceive variations from the extracellular setting and make a response. MAPK signaling cascades have been proven to be related to a diverse array of cellular programs which include cellular migration, proliferation, differentiation, and apoptosis [35,36]. Our recent publication has demonstrated that the MAPK signaling pathway can positively influence odontogenic differentiation of HDPCs [21,37,38]. Results of the present showed that icariin increased phosphorylation levels of JNK, p38, and ERK within 10 min of induction. In the collaborative experiment, expression levels of DSPP and DMP 1 protein, ALP activity, and the formation of calcium nodules could be partially suppressed by corresponding inhibitors, respectively. This is also consistent with previous research showing that icariin can promote osteoblast differentiation, proliferation, and mineralization by activating MAPK signaling [35,36]. Taken together, our present results provide direct evidence that icariin can enhance odontogenesis of HDPCs. As a promising agent, icariin provides new insights into vital pulp therapy. However, further in-depth investigation is a prerequisite to demonstrate precise signaling mechanisms and its therapeutic utility.

## 5. Conclusions

In conclusion, icariin is biocompatible. It can accelerate odontogenic differentiation and mineralization by triggering MAPK signaling pathway in HDPCs.

## Figures and Tables

**Figure 1 medicina-58-00434-f001:**
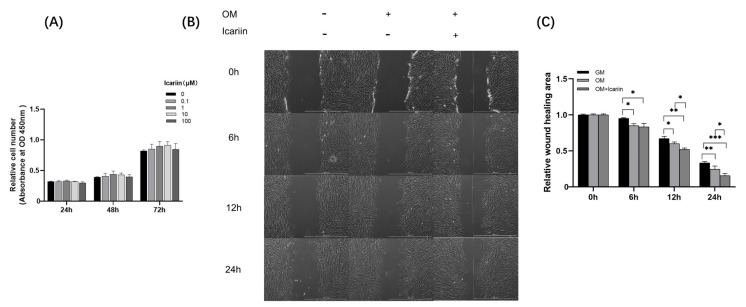
Effect of icariin on cytotoxicity and migration of HDPCs. (**A**) HDPCs were exposed to icariin at different concentrations for 24, 48, 72 h. Medium without icariin was used as a negative control (*p* > 0.05). (**B**) HDPCs were cultured with or without 10 μM icariin. Images of HDPCs migration were captured at 0, 6, 12, and 24 h (scale bar = 1000μM). (**C**) Quantitative analysis of wound migration area was normalized to GM. Bars show means ± standard deviation (* *p* < 0.05, ** *p* < 0.01, *** *p* < 0.001, HDPCs: human dental pulp cells, GM: growth medium, OM: odontogenic differentiation medium).

**Figure 2 medicina-58-00434-f002:**
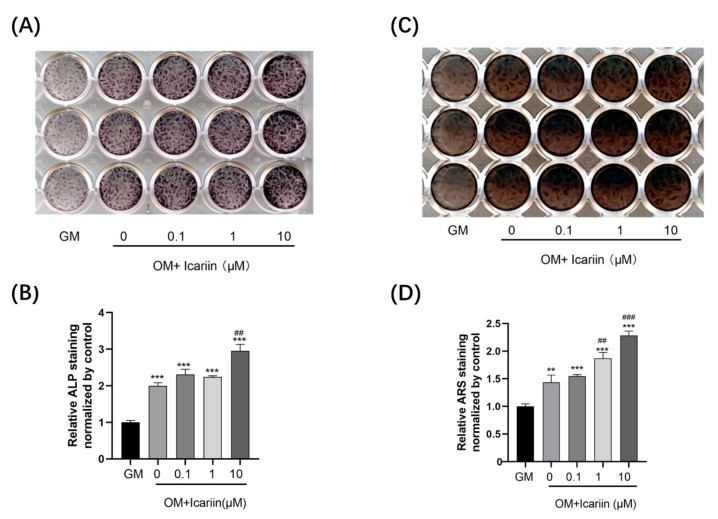
Mineralization of HDPCs after icariin treatment. (**A**,**B**) HDPCs were treated with different doses of icariin, displaying increased ALP activity at 10 days. (**C**,**D**) Treatment with icariin increased the capability of forming mineralized nodules on HDPCs at 14 days. Results were normalized to GM. Bars show means ± standard deviation (** *p* < 0.01 and *** *p* < 0.001 compared to GM; ## *p* < 0.01 and ### *p* < 0.001 compared to the OM without icariin treatment, ALP: alkaline phosphatase, ARS: alizarin red S).

**Figure 3 medicina-58-00434-f003:**
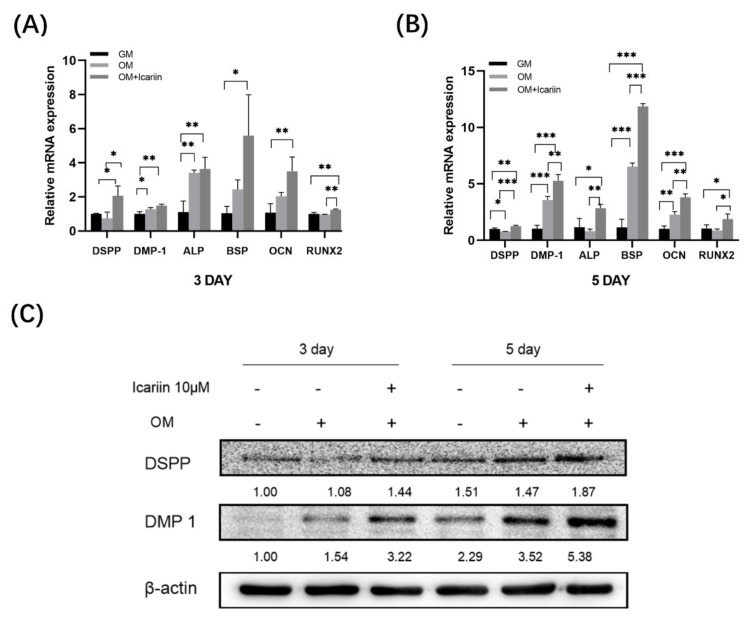
Effect of icariin on odontogenic differentiation of HDPCs. (**A**,**B**) Effect of icariin treatment on the expression of odontogenic differentiation markers at mRNA level. HDPCs were treated with 10 μM of icariin for 3 or 5 days. Expression levels of *DSPP*, *DMP 1*, *ALP*, *BSP*, *OCN*, and *RUNX 2* mRNAs were measured by quantitative real-time PCR. Gene expression was normalized against *GAPDH* (* *p* < 0.05, ** *p* < 0.01, *** *p* < 0.001). (**C**) DSPP and DMP 1 protein levels were upregulated after treatment with icariin at 3 or 5 days compared with both GM and OM. Bars show means ± standard deviation (*DMP 1*: dentin matrix protein 1, *DSPP*: dentin sialophosphoprotein, *RUNX2*: runt-related transcription factor 2, *OCN*: osteocalcin, *BSP*: bone sialoprotein and *GAPDH*: Glyceraldehyde-phosphate dehydrogenase).

**Figure 4 medicina-58-00434-f004:**
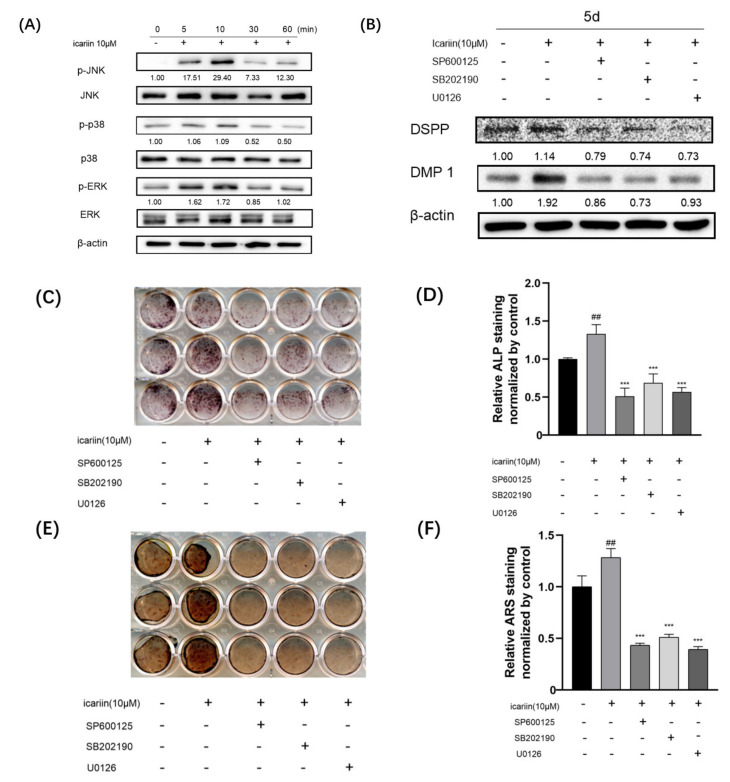
MAPK signaling pathway involved in icariin-mediated odontogenic differentiation of HDPCs. (**A**) Icariin activated MAPK pathways in HDPCs, JNK, p38, and ERK total proteins and phosphorylated proteins were assayed by western blot for the indicated time points (0, 5, 10, 30, and 60 min). (**B**) MAPK inhibitors (SP600125, SB202190, and U0126) suppressed expression levels of DSPP and DMP 1 proteins. (**C**,**D**) Icariin induced ALP activities were significantly blocked by the presence of MAPK inhibitors. (**E**,**F**) Icariin enhanced calcium deposition of HDPCs by mainly restraining MAPK inhibitors. Results were normalized to OM. Bars show means ± standard deviation (## *p* < 0.01 compared to OM; *** *p* < 0.001 compared to OM with icariin treatment). (MAPK: Mitogen-activated Protein Kinase, SP600125: JNK inhibitor, SB202190: p38 inhibitor, and U0126: ERK inhibitor).

**Table 1 medicina-58-00434-t001:** List of Primer Sequences included in this study for real-time PCR.

Genes	Primer Sequences (5′-3′)
*DMP 1*	Forward: TGG TCC CAG CAG TGA GTC CA
	Reverse: TGT GTG CGA GCT GTC CTC CT
*DSPP*	Forward: GGG AAT ATT GAG GGC TGG AA
	Reverse: TCA TTG TGA CCT GCA TCG CC
*RUNX2*	Forward: CCA GAT GGG ACT GTG GTT AC
	Reverse: ACT TGG TGC AGA GTT CAG GG
*ALP*	Forward: CGG GCA CCA TGA AGG AAA
	Reverse: GGC CAG ACC AAA GAT AGA GTT
*OCN*	Forward: CTC ACA CTC CTC GCC GTA TT
	Reverse: GCT CCC AGC CAT TGA TAC AG
*BSP*	Forward: AGC GAA GCA GAA GTG GAT GAA
	Reverse: CTG CAT TGG CTC CAG TGA CA
*GAPDH*	Forward: CAT CAC CAT CTT CCA GGA G
	Reverse: AGG CTG TTG TCA TAC TTC TC

## Data Availability

Data sharing is not applicable.

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
