# Peer review of "Odontogenic Effect of Icariin on the Human Dental Pulp Cells"

_medicina, 2022, doi:10.3390/medicina58030434_

Round 1

Reviewer 1 Report

  1. Why did the authors use these concentrations of icariin? Is it physiology or clinically relevant?
  2. I suggest to perform BrdU incorporation assays (detect DNA synthesis) to check cell viability of icariin.
  3. Fig 1A, Cell viability was assessed at only one time point 24 h, this does not provide a clear picture of the cell behavior. Please show results over a period of time.
  4. How did authors confirm that there were no genomic DNA contaminations in the isolated total RNA?
  5. line 66: Cells at passage at 3-5, authors can directly write passage 3-5

6. line 78: CO2 Not CO2 

Author Response

Reviewer Comments:

Comment 1: Why did the authors use these concentrations of icariin? Is it physiology or clinically relevant?

Authors’ response: The authors are very thankful for this comment. In the previous study, 0.01-160 μM icariin was chosen to induce the migration and osteogenic differentiation in different cell types [11,12,14]. In our paper, 0.1-100 μM icariin did not show the cytotoxicity of human dental pulp cells. In the alizarin red staining and alkaline phosphatase staining, 0.1, 1, 10 μM induced the odontogenesis function. Therefore, we chose the 10 μM for the further mechanism analysis.

Comment 2:  I suggest to perform BrdU incorporation assays (detect DNA synthesis) to check cell viability of icariin.

Authors’ response: Thank you for your comments. BrdU incorporation assay is a non-isotopic immunoassay for quantification of BrdU incorporation into newly synthesized DNA of actively proliferating cells. It is sensitive, rapid, and easy to perform. The WST-1 assay also provides a sensitive means for performing a quantitative cell viability assay. However, if the BrdU incorporation assay is necessary, we might need more time to order the kit and perform it.

Comment 3: Fig 1A, Cell viability was assessed at only one time point 24 h, this does not provide a clear picture of the cell behavior. Please show results over a period of time.

Authors’ response: According to the suggestion, the authors repeated the cell viability assay and add two time point (48 h and 72 h) which have been modified in the revised manuscript (Figure 1).

Comment 4: How did authors confirm that there were no genomic DNA contaminations in the isolated total RNA?

Authors’ response: Thank you for your comments. Firstly, RNAs extraction was performed with Trizol following the procedures supplied by manufacturer strictly, which can avoid DNA contamination. Secondly, A260 /A280 ratio was used to assess RNA purity. In our paper, the A260/A280 ratio was set at 1.9–2.1, which can exclude the DNA contamination.

Comment 5: line 66: Cells at passage at 3-5, authors can directly write passage 3-5.

Authors’ response: As suggested, the authors have modified the section (sentences by highlighting)。

Comment 6: line 78: CO2 Not CO2

Authors’ response: As suggested, the authors have modified in the manuscript (sentences by highlighting)。

Reviewer 2 Report

The authors claimed that icariin could up-regulate odontogenic differentiation of HDPCs by triggering the MAPK signaling pathway. However, there are several concerns in this manuscript.

  1. Only ALP and ARS staining with gene and protein expression profile are not enough to show odontogenic differentiation. They need to show the effect of icariin on the functionality of odontogenic differentiation.
  2. All the protein expression should be quantified from multiple biological experiments.
  3. In comparison of Figure 2B/2D and Figure 4D/4F, the effect of Icaarin (10 µM) on ALP and ARS are quite different (much lower in Figure 4D and 4F), they should explain the difference and repeat the experiments if needed.

Author Response

Reviewer Comments:

Comment 1: Only ALP and ARS staining with gene and protein expression profile are not enough to show odontogenic differentiation. They need to show the effect of icariin on the functionality of odontogenic differentiation.

Authors’ response: Thank you for your comments. odontogenesis and osteogenesis share similar dynamic procedure and mechanism in the process of mineral phase. Studies have indicated that ALP staining and ARS staining were effective methods to assess the result of inducing osteogenic and odontogenic differentiation [8,18,26]. DSPP and DMP-1 were two classical markers of odontogenic differentiation for human dental pulp cells (HDPCs) [9,27-29]. In our paper, ARS staining, ALP staining and the expression level of DSPP, DMP1, RUNX2, ALP, OCN, and BSP were used to verify the effect of icariin in inducing odontogenic differentiation for HDPCs.

Comment 2: All the protein expression should be quantified from multiple biological experiments.

Authors’ response: According to the suggestion, the authors quantified the protein expression which have been modified in the revised manuscript (Figure 3 and 4).

Comment 3: In comparison of Figure 2B/2D and Figure 4D/4F, the effect of Icariin (10 µM) on ALP and ARS are quite different (much lower in Figure 4D and 4F), they should explain the difference and repeat the experiments if needed.

Authors’ response: Thank you for your comments. In Figure4D/4F, we did not set the negative control group when compared with Figure 2B/2D. If we compared the Icariin (10 µM) group and positive control group in Figure 2B/2D and Figure 4D/4F which was attached below, the effect of Icariin (10 µM) was not quite different.           And the difference might be result from the HDPCs that were isolated from different donors.

Round 2

Reviewer 2 Report

The authors revised the manuscript sufficiently.